# Lung Ultrasound in Children with Cystic Fibrosis in Comparison with Chest Computed Tomography: A Feasibility Study

**DOI:** 10.3390/diagnostics12020376

**Published:** 2022-02-01

**Authors:** Ioana Mihaiela Ciuca, Liviu Laurentiu Pop, Mihaela Dediu, Emil Robert Stoicescu, Monica Steluta Marc, Aniko Maria Manea, Diana Luminita Manolescu

**Affiliations:** 1Pediatric Department, “Victor Babes” University of Medicine and Pharmacy Timisoara, Eftimie Murgu Square No. 2, 300041 Timisoara, Romania; ciuca.ioana@umft.ro (I.M.C.); liviupop@umft.ro (L.L.P.); dediu.mihaela@umft.ro (M.D.); 2Radiology Department, “Victor Babes” University of Medicine and Pharmacy Timisoara, Eftimie Murgu Square No. 2, 300041 Timisoara, Romania; stoicescu.emil@umft.ro (E.R.S.); dmanolescu@umft.ro (D.L.M.); 3Research Center for Pharmaco-Toxicological Evaluations, “Victor Babes” University of Medicine and Pharmacy Timisoara, Eftimie Murgu Square No. 2, 300041 Timisoara, Romania; 4Pulmonology Department, “Victor Babes” University of Medicine and Pharmacy Timisoara, Eftimie Murgu Square No. 2, 300041 Timisoara, Romania; 5Neonatology and Puericulture Department, “Victor Babes” University of Medicine and Pharmacy, Eftimie Murgu Square No. 2, 300041 Timisoara, Romania; manea.aniko@umft.ro

**Keywords:** lung ultrasound, cystic fibrosis, computed tomography comparison, CT

## Abstract

Background: Cystic fibrosis (CF) lung disease determines the outcome of this condition. For lung evaluation processes, computed tomography (CT) is the gold standard, but also causes irradiation. Lately, lung ultrasound (LUS) has proven to be reliable for the diagnosis of consolidations, atelectasis, and/or bronchiectasis. The aim of our study was to evaluate the value of a newly conceived LUS score by comparing it to the modified Bhalla CT score. A further aim was to evaluate the correlation between the score and the lung clearance index (LCI). Methods: Patients with CF were screened by LUS, followed by a CT scan. Spearman’s test was used for correlations. Results: A total of 98 patients with CF were screened, and 57 were included in the study; their mean age was 11.8 ± 5.5 (mean ± SD) years. The mean LUS score was 5.88 ± 5.4 SD. The LUS CF score had a very strong correlation with the CT score of rs = 0.87 (*p* = 0.000). LUS showed a good sensibility for detecting atelectasis (Se = 83.7%) and consolidations (Se = 94.4%). A lower Se (77.7%) and Sp (9%) were found for cylindrical bronchiectasis. Conclusion: Our study shows that LUS and the lung CF score are parameters that can be used with a complementary role in the diagnosis and monitoring of CF lung disease in children.

## 1. Introduction

Cystic fibrosis is a complex disease characterized by significant clinical polymorphism, a special evolution, and severe complications that raise problems in the individual monitoring and management of the disease [1]. The pulmonary condition remains the most important issue that dictates the prognosis of the disease [2]. Therefore, an early diagnosis of pulmonary complications and the preservation of the lung function are essential. For the diagnosis and monitoring of CF lung disease, many investigations are used, from widely accessible chest X-ray examinations (CXR) [3]—which have lower sensitivity—to computed tomography (CT), the current gold standard [4]. HRCT is very sensitive for the detection of any structural changes, but its repeatability is restricted because of its significant irradiating potential. On the other hand, lung ultrasound is a non-irradiating, easy-to-use method, reliable for the detection of severe childhood pulmonary diseases, from pneumonia [5] to pulmonary fibrosis [6]; therefore, it is important to evaluate its potential in the accurate detection of CF lung disease.

Presently, lung ultrasound has demonstrated efficiency in the diagnosis of child pneumonia [5,7] and other frequent childhood diseases, such as bronchiolitis, pneumothorax, atelectasis [8], pleural effusion, and pulmonary contusion [9]. It is important to note that lung ultrasound is more sensitive for the detection of smaller lesions [10]. LUS is also valuable for the examination of children diagnosed with tuberculosis, as it seems to be more sensitive than CXR [11]—especially for sickle cell disease [12], or rare diseases such as NEHI [13].

Studies have shown that LUS value in CF exacerbations [14] correlates with lung function tests [15], and reliable correlations between LUS and Chrispin–Norman X-ray score [16] or modified Bhalla CT score have been published [17].

The progression of structural lung deterioration requires objective measurements, such as CT scores [18,19,20,21], which offer the necessary support for correct monitoring and the possibility of a follow-up—a mandatory procedure for accurate lung evaluation.

Numerous lung ultrasound scores were developed as consistent, non-invasive tools for numerous diagnoses, such as respiratory COVID-19 in adults [22,23,24], pneumonia in elders [25], ARDS [26], and lung recruitment ventilation [27,28], with significant practical results. Similarly, for children’s respiratory pathology, important achievements have been made by the use of LUS scores to evaluate COVID-19 pneumonia in neonates, pneumonia in children [29], and to predict the need for surfactants in neonates [30], or ventilation requirements in neonatal respiratory distress syndrome [31].

Therefore, in this study, we aimed to assess the value of a newly conceived lung ultrasound (LUS) score by comparing it to the HRCT modified Bhalla score, and to evaluate the correlation between the LUS score and the lung function expressed by the lung clearance index (LCI), which is the most accurate parameter for CF lung function evaluation [32,33].

The LUS scoring system is useful for the detection of multiple respiratory diseases, including features that can be found in CF, in addition to pneumonia [34] or COVID-19 [24].

Chest CT is the gold standard for the structural evaluation of CF lung disease, and the need of an objective marker led to creation of CT scores, which are able to estimate the severity and degree of the specific features that appear in CF [33]. The modified Bhalla score is an accurate and feasible way to assess the severity of CF—a lung parenchymal disease—as it is closely correlated with lung functions [33], severe genotype, and chronic Pseudomonas infections [32,33,35]. In the study conducted by Leung A. et al. [35], modified Bhalla score included the evaluation of the presence, severity, and extent of bronchiectasis, bronchial wall thickening, mucus plugging, atelectasis/consolidation, and air trapping, using a 0–3 severity scale [35], thus simplifying the original Bhalla scale [20].

Even CT scores are used as a surrogate outcome in the evaluation of cystic fibrosis lung disease. In terms of sensitivity, it seems that LCI is comparable with CT. LCI is an indicator of irregular ventilation distribution that sensibly detects abnormal lung structure and early changes in the lungs with CF [32,33]; its practical usefulness lies in its applicability in younger children who cannot be subjected to spirometry, as only tidal breathing is necessary for performing multiple-breath washout (MBW)—the method through which LCI is obtained [32]. Several studies have demonstrated that LCI is better correlated with CT scores than spirometry parameters, and is very sensitive for the detection of early changes in CF lungs [32,33], suggesting that “LCI may be even more sensitive than HRCT scanning for detecting lung involvement in CF” [32].

Therefore, we conclude that the comparison between our new LUS-CF score and a CT examination—the gold standard for structural CF changes—and LCI—the most accurate lung function parameter—is the right premise for this feasibility study.

## 2. Materials and Methods

### 2.1. Study Population

The study population was aged between 6 months and 18 years, diagnosed with typical cystic fibrosis and monitored at our CF center. They were invited to participate in the study, starting from October 2016 until March 2020. The study was approved by the Ethics Committee of the Clinical County Hospital (no.8/2016).

Each parent and, in cases over 12 years old, each child, signed the informed consent agreement regarding the agreement to participate in the study, in accordance with the Declaration of Helsinki.

### 2.2. Study Protocol

#### 2.2.1. Lung Ultrasound

LUS was performed at the first clinical evaluation, before the biological tests and the computer scanning.

We used an Alpinion E-CUBE 9 ultrasound system, scanning with a linear probe of 7–12 MHz frequency and a 3.5–5 MHz convex probe, corresponding to the thoracic wall dimensions. The lung ultrasound was performed by a pediatric pulmonologist with 7 years of experience in LUS, blinded to previous CT examinations, and the stored images were checked by a senior radiologist with 8 years of expertise in lung ultrasound.

Most of the compliant children were evaluated in supine and prone positions, and then in their mothers’ arms or in an upright position, if pleurisy was detected, for volume estimation.

The ultrasound evaluation protocol included scanning of the lung areas by longitudinal sections: right and left parasternal, medio-clavicular, anterior and posterior axillary, posterior by paravertebral, medio-scapular and posterior axillar lines. Moreover, the probe transversally scanned each intercostal space, in addition to the transabdominal approach through the liver and spleen window for costal diaphragmatic angles and retrocardiac consolidations. Separately, the hemithorax was virtually divided into 6 areas: 2 anterior—anterosuperior and anteroinferior; 2 lateral—superior and inferior lateral; and 2 posterior—superior and inferior [6,34]. The splenic ultrasound window was also used to evaluate the lower lobes of the left lung and the left costodiaphragmatic angle, as well as the hepatic window for the lower right lung artefacts.

LUS-CF scores were quantified as normal (0–1), mild (2–6), moderate (6–10), or severe (>10).

#### 2.2.2. CT

After the LUS examinations were performed, the patients underwent a CT scan every two years, as part of their regular evaluation, according to our national standards.

The CT scans were performed with a Philips MX 16 EVO 16-slice CT with dedicated pediatric protocols and a Neusoft NeuViz 16 Essence 16-slice CT in the Radiology Department of “Pius Brînzeu” County Emergency Clinical Hospital, Timisoara. The CT scans were optimally performed at 120 kVp, according to international standards. CT scans were acquired at a 2.5 mm and 5 mm slice thickness, with reconstruction images at 1.25 mm. The images were stored in the workstation, the hospital’s PACS system, and on CDs, and were interpreted by an experienced radiologist with CT competence and 16 years of experience.

The CT images were analyzed and scored using a modified Bhalla cystic fibrosis score for HRCT [35]. For evaluating the lung lesions, the score considered the quantification of the injuries to the respiratory tract: type (cylindrical, varicose, and saccular) and extension of bronchiectasis, the thickening of the bronchial walls in different stages (mild, moderate, and severe) and the extent of mucus plugging. The quantification of air-trapping zones and the extension of the lung parenchymal lesions (a consolidation area with air bronchogram, consolidation zones, and atelectasis) were also examined.

The scores were achieved for all five lung regions corresponding to the right upper, middle, and lower lung zones, as well as the left upper and left lower lung zones. The final lesion score was obtained by summing up the five-lobe score. According to the total severity score, CT scores were classified as mild (0–33), moderate (34–66), or severe (> 66) [35].

#### 2.2.3. Lung Function

Spirometry was performed in all patients over the age of 5 years old, as part of the biannual or 3-monthly evaluation, according to their age and infection status. The standards imposed by the American Thoracic Society/European Respiratory Society [36], along with the Global Lung Function Initiative 2012 reference equations [37], were used to calculate the percentage of the predicted parameter values, using a CareFusion machine.

LCI obtained by tidal breathing and by multiple-breath nitrogen (N2) washout was determined using Quark PFT (COSMED, Italy). The LCI was calculated as the number of lung volume turnovers (i.e., the cumulative expired volume divided by the functional residual capacity) needed to lower the end-tidal tracer gas concentration below 2.5% (1/40 of starting level), with the normal values considered below 7 [38].

### 2.3. Statistical Analysis

For the descriptive statistics, the percentage values for categorical variables and continuous variables were expressed as the mean ± standard deviation. The Shapiro–Wilk test was used to establish the distribution of our quantitative data. The data were investigated using IBM SPSS Statistics version 26.0. Spearman’s correlation coefficient was used for the evaluation of the relationships between the quantitative variables. The coefficient Spearman’s rho < 0.2 was significant for showing the lack of relationship between the variables. The correlation was considered weak if Spearman’s rho was between 0.2 and 0.29, moderate with Spearman’s rho 0.3–0.39, strong relationship if Spearman’s rho had a value in the 0.4–0.67 interval, and very strong when Spearman’s rho > 0.7 [39]. The Kruskal–Wallis test was applied to compare the medians between the groups. The specificity and sensitivity rates were calculated, in addition to the positive and negative predictive values, while *p*-values were considered significant if *p* < 0.005.

## 3. Results

### 3.1. Descriptive Data

A total of 98 patients with CF were screened, and 57 were included in the study, as CT was performed at their biannual evaluation. Their mean age was 11.8 ± 5.5 SD years (ranging between 3.3 and 21.8 years old), and 42.1% were females. Most of the patients had a severe genotype, almost half of them (49.1%) being f508 del homozygous; the f 508 del allele was present in 71.05% of the cases, followed by G542X in 6.1%. The percentage of chronically infected patients was = 49.12% (33.3% with *Pseudomonas aeruginosa* 15.7% with *Staphylococcus* strains, and 7% were polymicrobial).

### 3.2. LUS CF Score

The artefacts used to define the pathological elements were as follows (Table 1): the presence of A lines—normal aspect = 0 points; less than 3 B lines, thin (< 2 mm in width)/intercostal space = 0 points; more than 3 distinctive B lines or 1 coalescent B line = 1 point, quantifying interstitial inflammation or small bronchiectasis (Figure 1) confirmed by CT (Figure 2); more than 2 coalescent B lines = 2 points, suggestive of alveolo-interstitial inflammation or mucus plugging with loss of aeration; either bronchial wall thickening or subpleural consolidation < 1 cm = 3 points, associated with the absence of A lines quantified either as small atelectasis or cystic bronchiectasis with mucus plugging; subpleural lung consolidation > 1 cm, without bronchogram = 4 points; quantified atelectasis (Figure 3)/consolidation with bronchogram = 5 points.

Subpleural consolidation were detected by CT scan (Figure 4) in 33.3% of patients, and confirmed by LUS in 31.5% (Figure 3).

The right hemithorax of the same patients revealed the presence of B lines for cylindrical bronchiectasis via LUS (Figure 5).

The calculation of the score was done by summing up the lesions detected in the six zones of every corresponding hemithorax. The mean LUS score was 5.88 ± 5.4 SD, ranging from 0–21. The mean CT score was 38.14 ± 11.1, consistent with the moderate structural lung damage, ranging from 4 points to a maximum of 82 points.

### 3.3. Spearman’s Correlation Test

Taking into consideration the fact that our data have a nonparametric distribution, we used Spearman’s rho coefficient in order to evaluate the correlation between LUS score and CT score, FEV1, FEF 25–75, and LCI.

#### 3.3.1. LUS-CF Score and CT Score

The LUS-CF score had a very strong correlation with the CT score of rs = 0.87, showing important statistical significance (*p* = 0.000), suggesting a good reliability of the LUS-CF score in the evaluation of CF lung parenchymal deterioration.

We divided the patients according to their CT score: mild disease (0–33), moderate disease (34–66), and severe disease (> 67) (Figure 6). The correlation in patients with mild disease, expressed by an LUS-CF score from 0 to 7, was weak (rs = 0.439) (*p* = 0.014), while in patients with moderate disease, the correlation coefficient strongly increased to rs = 0.57, with good statistical significance (*p* = 0.01). By applying the Kruskal–Wallis test, we found a statistical difference between the median LUS values in the categories of the CT score (H = 39.845, *p* = 0.000; Table 2).

In patients with important structural lung damage (Figure 7), quantified as severe disease, expressed by a CT score > 66, the correlation was strong (rs = 0.83), with statistical significance (*p* = 0.002).

#### 3.3.2. LUS-CF Score with Lung Function Parameters

Evaluating the relationship between the LUS score and LCI, the Spearman’s correlation coefficient rs = 0.8 revealed a strong, statistically significant correlation (*p* = 0.000)—an encouraging significant association between the structural lung disease and lung function. Additionally, the relationship between the LUS-CF score and the spirometry parameters was evaluated, and a strong negative correlation was found with FEV1 rs = −0.65 (*p* = 0.000) and FEF 25–75 rs = −0.542 (*p* = 0.000).

### 3.4. LUS Sensitivity and Specificity

The assessment of LUS sensitivity and specificity in bronchiectasis detection varied with the form of bronchiectasis: for cylindrical bronchiectasis(Figure 8A,B), LUS Se = 77.7%, Sp = 9%, PPV = 80.7%, and NPV = 76.9%, while for saccular bronchiectasis (Figure 7), a moderate Se = 68.4%, with good Sp = 94.9%, PPV = 88.8%, and NPV = 94.7% were found.

As for varicose bronchiectasis (Figure 9 A,B), a very low Sp = 25% and NPV = 16.6% were calculated, with a satisfactory PPV = 88.8% and Se = 68.4%.

The results regarding atelectasis (Figure 10 and Figure 11) and consolidation detection were significant. LUS showed good sensitivity and specificity in detecting atelectasis (Figure 10)—Se = 83.7%, Sp = 94.5%, PPV = 92.5%, NPV = 72.3%—and consolidations—Se = 94.4%, Sp = 93.02%, PPV = 89.4%, NPV = 97.3%.

As for bronchial thickening, low sensitivity and specificity were found: Se = 31.7%, Sp = 35.2%, PPV = 54.1, NPV = 14.2%.

We could not calculate the reliability of LUS for air trapping or for mucus plugging because of a lack of specific artefacts.

## 4. Discussion

The literature on LUS-CF is limited, as CF is a rare disease with few patients, and a small number of specialists in LUS.

In this study, we found that the LUS-CF score is a valuable instrument not only to reveal the presence and quantification of parenchymal injury in CF, but also for expressing the relationship with lung clearance index—the most accurate CF functional parameter.

LUS can show many ultrasound abnormalities, such as B lines, pleural line abnormalities, important consolidations, and atelectasis. In addition to its diagnostic practicality, LUS also seems to be effective for detection of exacerbations [40], showing good correlation with lung function [17,41]. Few studies have investigated the significance of LUS in CF [15,16,17,40,41], but emerging evidence remains to be shown. Our study is the first to describe the LUS artefacts corresponding to CT lung lesions, quantifying all lung injuries potentially detected by LUS. Furthermore, this is the first study to evaluate the relationship between lung structural damage expressed by LUS scoring and functional issues expressed by LCI, as we previously noted this correlation between structure and function [41].

The first study that presented the CF lung ultrasound artefacts in cystic fibrosis described the presence of interstitial syndrome, bronchiectasis, alveolar consolidation, and pleural signs, but was published only in abstract, by our group [41]. Strzelczuk-Judka subsequently reported the CF-USS (cystic fibrosis ultrasound score), which evaluated the presence and extent of pleural irregularities, focal or coalescent ”lung rockets” B lines, subpleural consolidations, and pleural fluid, showing a positive correlation with Chrispin–Norman CXR scoring systems (r = 0.52, *p* = 0.0002) [16]. As in our study, they acknowledged an important limitation in the inability to visualize the respiratory airway deterioration, e.g., bronchiectasis and mucus plugs.

Peixoto et al. reported a score that included A pattern and B pattern and C pattern (consolidation), stratifying patients into A profile, B profile, C profile, or mixed profiles compared to CT, and reported a good correlation between LUS and CT [17]. The presence of pleural effusion was not scored (declared as very rare), nor were the pleural irregularities (only describing the finding) [17], similar to our study. The choice of excluding pleural effusion from our LUS-CF score was based on the fact that pleural effusion is not a specific feature of CF.

Hassanzad et al. noted in LUS the presence of pleural thickening, atelectasis, air bronchogram, B lines, and consolidation, compared the findings with corresponding CXR and HRCT, and evaluated the diagnostic performance of LUS and CXR for every artefact, with satisfactory results [40]. Similar to our discoveries, a good diagnostic performance for the detection of consolidation was reported in this paper.

Regarding pleural irregularities, Strzelczuk-Judka noted pleural irregularities similar to bronchiolitis in one patient [16]; Peixoto described this feature, but did not quantify it [17]. We did not take into consideration the presence of the pleural irregularities for this 2016 starting study, because CT did not show a specific corresponding finding; therefore, we considered it normal appearance at the time of our LUS-CF score’s development. Neither of the others studies stated that pleural irregularities or thickening would correspond to a specific modification in CT.

With previous experience of our group on LUS in CF [41], we noted that LUS artefacts may quantify different pathological expression. As exemplified in the results of the present paper, B lines can also quantify interstitial inflammation or small bronchiectasis, as confirmed by CT in our study. Similarly, the coalescent B lines can be suggestive of alveolo-interstitial inflammation or mucus plugging with loss of aeration, or bronchial wall thickening. We observed that subpleural consolidation with absence of A lines quantified small atelectasis, but also cystic bronchiectasis filled with mucus. The lack of specificity for mentioned artefacts led us to be cautious in asserting the specificity of LUS for CF lung disease. However, the LUS-CF score showed a very good correlation with CT, as mentioned, and was highly sensitive in detecting parenchymal abnormalities. Other studies suggest the role of LUS in CF exacerbations [42], showing its superiority to CXR in terms of the detection of consolidations, pleural effusion and irregularities compared to CT examination [40] in exacerbations of CF in patients. In previously published papers, a good correlation was found between the LUS and Chrispin–Norman CXR score [16], and also with the modified Bhalla CT score [17], which is also reflected in our own findings.

Our findings reveal a very strong positive linear correlation between the LUS-CF score and LCI (rs = 0.87, *p* = 0.000), suggesting the reliability of the LUS-CF score in the detection of functional impairment in CF patients. The correlation coefficient between LUS and LCI was superior to the correlation with spirometry parameters, which can be explained by the increased sensitivity of LCI in the detection of lung function impairment and the specificity of the LUS-CF score that included all B line spectra, consolidations with or without bronchiectasis, and loss of aeration.

A strong negative correlation was found with CF-specific parameters for obstructions such as FEV1 (rs = −0.65 (*p* = 0.000) and FEF 25–75 (rs = −0.542 (*p* = 0.000). These findings are similar to those of another study that evaluated LUS patterns in CF, and which showed a correlation with lung function expressed by pre-BD FVC r = 0.538 and pre-BD FEV1 r = 0.536 [17].

Descriptive data on LUS artefacts in a number of conditions have been published, but the development of LUS scores has opened the door to objective evaluation. These studies show a correlation between LUS scores and inflammation [29], lung function in CF, and even mortality prognosis [43] in several diseases.

The reliability of LUS in CF lung disease is sustained by its very good sensitivity and specificity in detecting consolidations (Se = 94.4%, Sp = 93.02%), a reliable positive predictive value of 89.4%, and an important NPV = 97.3%. LUS detected atelectasis with a good sensitivity of 83.7% and a significant specificity of 94.5%. The positive predictive value was important for atelectasis (PPV = 92.5%), and NPV = 72.3%. These findings are similar to those of other studies [16,17,40].

As for LUS sensitivity and sensibility in bronchiectasis, detection varied with the type of bronchiectasis, with a decent sensitivity for detecting cylindrical bronchiectasis (77.7%) and saccular bronchiectasis (Se = 68.4%, Sp = 94.9%), but a lower specificity of 9% for cylindrical bronchiectasis, explained by the fact that they were quantified by B lines which, as stated, suggest several diseases [44].

The results regarding atelectasis detection were significant; LUS showed good sensibility and specificity in detecting atelectasis: Se = 83.7%, Sp = 94.5%, PPV = 92.5%, NPV = 72.3%. As for bronchial thickening, low sensitivity and specificity were found: Se = 31.7%, Sp = 35.2%, PPV = 54.1, NPV = 14.2%.

This study has a few limitations. Lung ultrasound was performed by a single practitioner, and while the number of patients included in the study was satisfactory, it is possible that a larger study would have offered more consistency. Another limitation is that important artefacts such as air trapping/emphysema—one of the premature signs of lung damage in CF—has no US correspondence. No artefact for emphysema quantification was found, nor has any been published previously. Furthermore, small mucus plugging or mild bronchial wall thickening was not detected by LUS, as no LUS artefacts for these changes have been identified to date.

An additional important issue is related to the lack of specificity of B lines, which can quantify a number of diseases [44], ranging from interstitial lung disease or bronchiolitis to acute respiratory distress syndrome or bronchiectasis [45]. We specifically quantified different extensions of bronchiectasis (i.e., cylindrical, varicose, cystic) by the presence of B lines to increase the accuracy of our study, and we eliminated the patients in exacerbations in order to decrease the misinterpretations of interstitial inflammation that can occur in exacerbations with bronchiectasis. As bronchiectasis is the hallmark of CF lung disease, special attention was given to its detection via LUS, and we found LUS to be reliable for the detection of bronchiectasis—a finding that is similar to those of recent studies in adult pathology [46]. The presence of more than three individual B lines per intercostal space, in longitudinal sections, detected before CT examination, suggested certain structural changes. In some of the patients, the above-mentioned studies quantified bronchiectasis to different degrees; in others, bronchial wall thickening or small mucus plugging—specific features identified on CT; therefore, they are not specific to a single lesion, but their presence definitely indicates a lung lesion.

The validation of our score is expressed by the strong correlation found with the modified Bhalla CT score—especially in severely altered lung diseases. However, CT remains the gold standard for CF lung morphological evaluation, as it can accurately detect alterations that are not encryptable via LUS so far, such as air trapping and mucus plugging, considering that bronchiectasis and air trapping are important validated outcome parameters [47].

Even if HRCT is the gold standard for lung disease in CF, it cannot be used as often as necessary—especially in children diagnosed with CF, who require sedation when being exposed to repetitive irradiating scans, even if a low dosage is used.

LUS can be a reliable instrument for screening and monitoring children with CF—especially in the advanced forms of the disease—reducing the levels of exposure to CT radiation.

The strength of this study is the presentation of the LUS-CF score as a non-invasive and reliable tool for the screening of advanced structural damage in CF.

## 5. Conclusions

LUS in the monitoring of CF patients’ lung disease is reliable for advanced lung disease and for lesions of moderate severity, but for the detection of early changes LUS is not a consistent method of lung investigation, with CT remaining the gold standard.

Our study has shown that the LUS-CF score is a parameter that can be used for an associated evaluation, and can play a complementary role in the diagnosis and monitoring of CF lung disease in children.

## Figures and Tables

**Figure 1 diagnostics-12-00376-f001:**
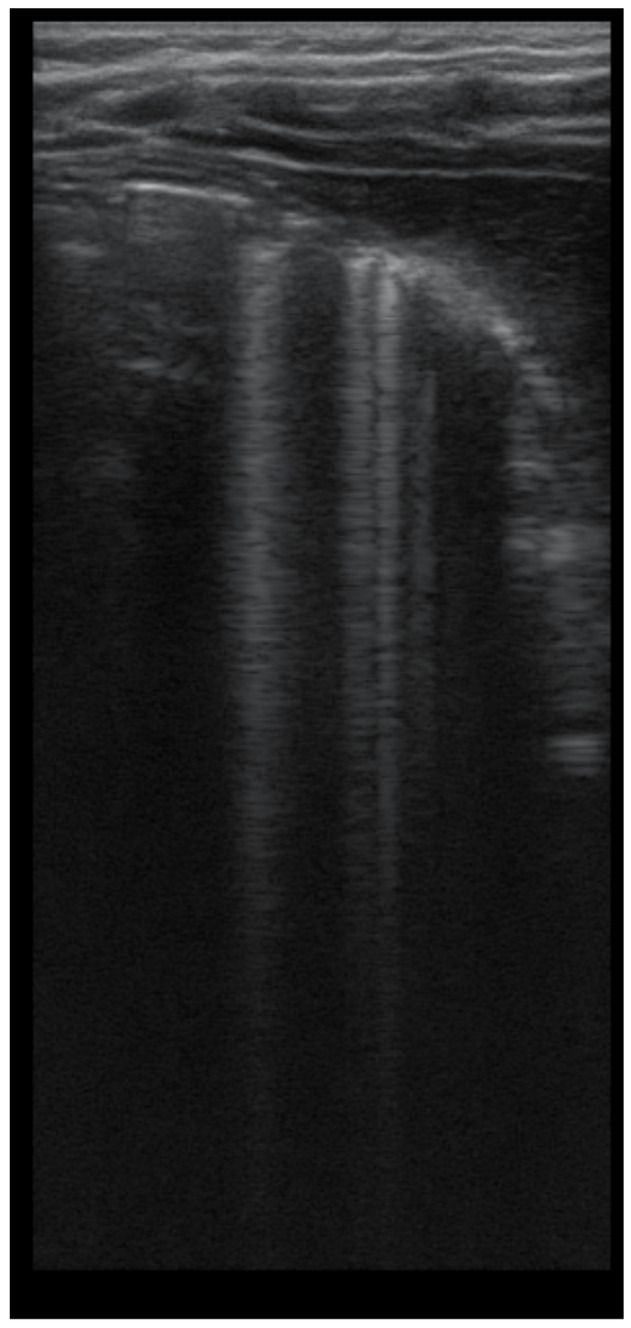
LUS image shows B lines > 3, LUS score = 2. The corresponding CT image (Figure 2) shows bronchiectasis.

**Figure 2 diagnostics-12-00376-f002:**
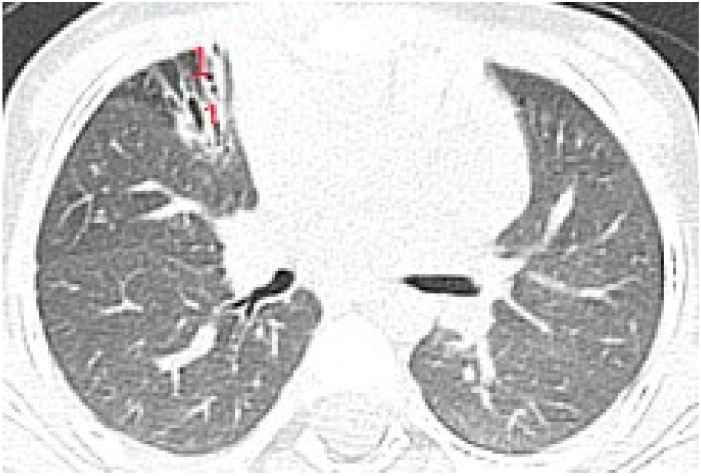
CT reveals (1) peripheral cylindrical bronchiectasis with mucus plugging.

**Figure 3 diagnostics-12-00376-f003:**
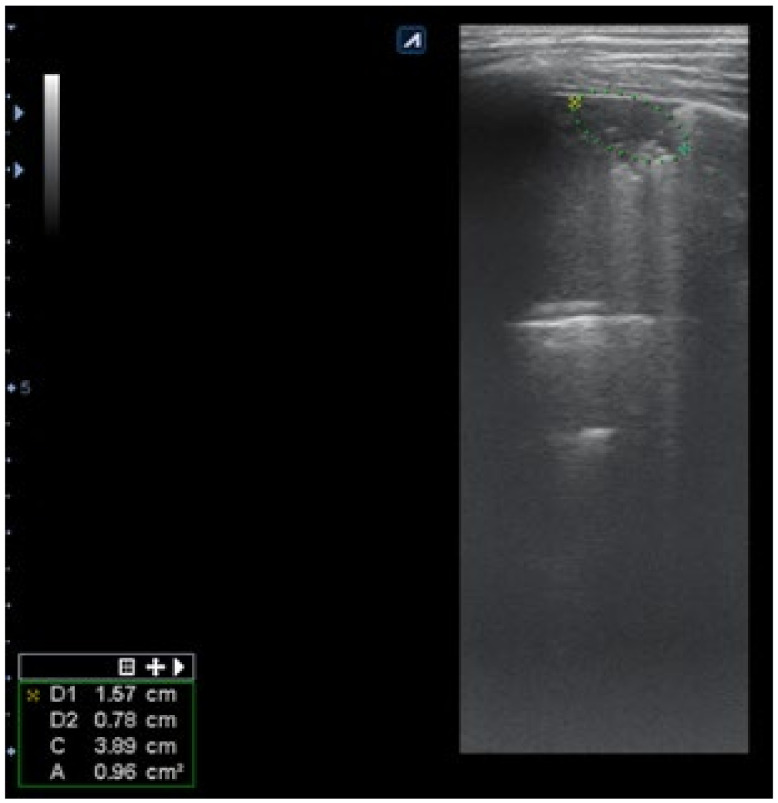
LUS: subpleural consolidation of 1.75 cm/0.78 cm, surface of 0.96 cm^2^ (score 4), without bronchogram, with adjacent B lines (2 points); A lines also present, examination of left posterior hemithorax.

**Figure 4 diagnostics-12-00376-f004:**
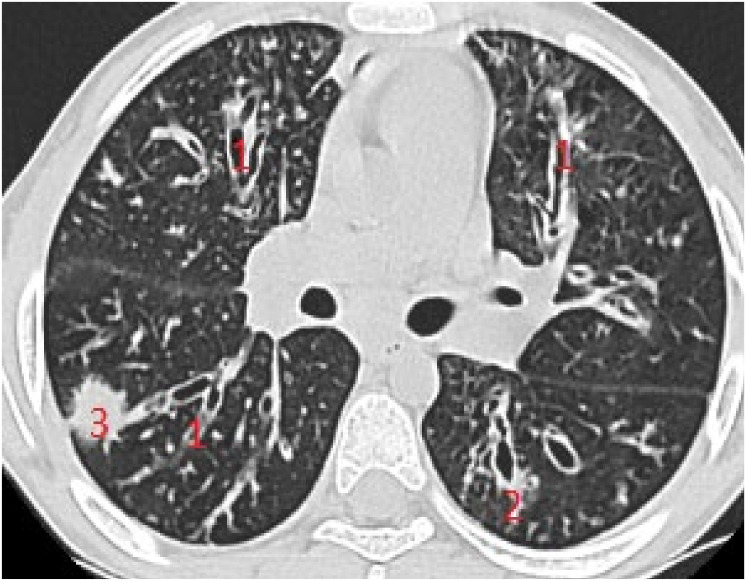
CT scan of the same patient, with various types of bronchiectasis: (1) cylindrical bronchiectasis with moderate bronchial wall thickening; (2) varicose bronchiectasis; and (3) a round/spiculated consolidation, corresponding to previous LUS consolidation. CT score = 62.

**Figure 5 diagnostics-12-00376-f005:**
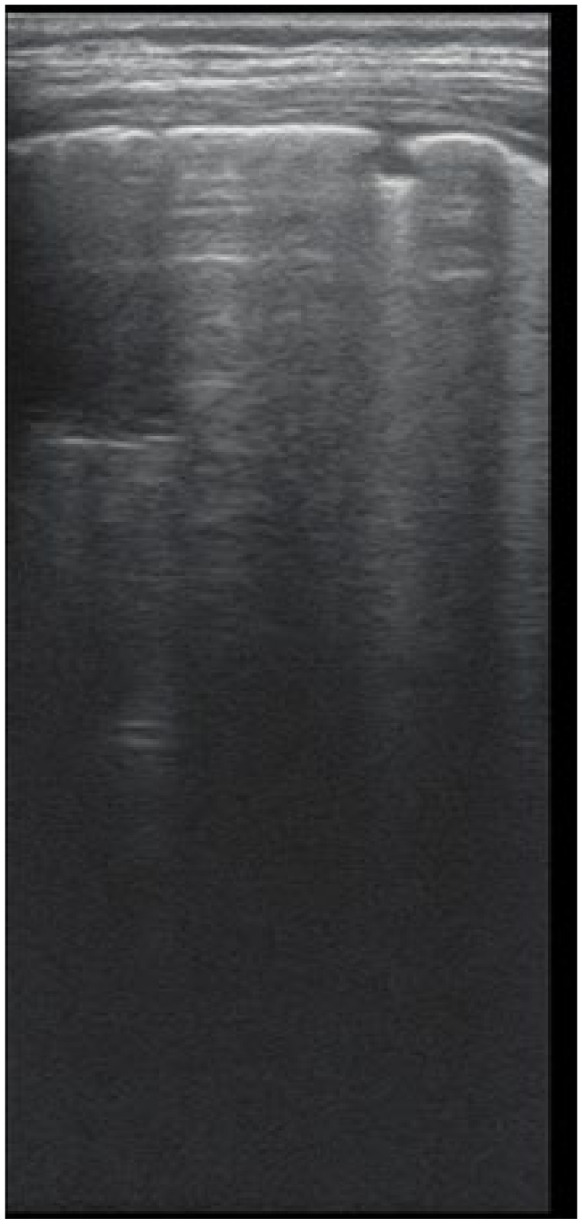
LUS: coalescent B lines, with a very small subpleural consolidation (2 points) and 2 coalescent B lines (2 points), corresponding to mucus-filled varicose bronchiectasis; examination of the same patient’s right posterior hemithorax.

**Figure 6 diagnostics-12-00376-f006:**
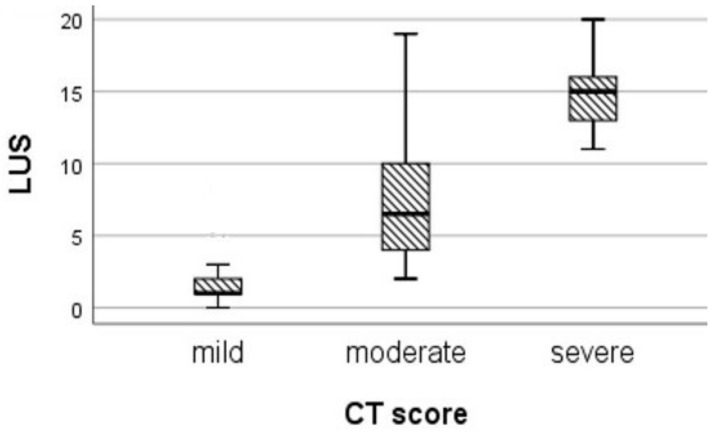
Median LUS scores in patients classified by CT score.

**Figure 7 diagnostics-12-00376-f007:**
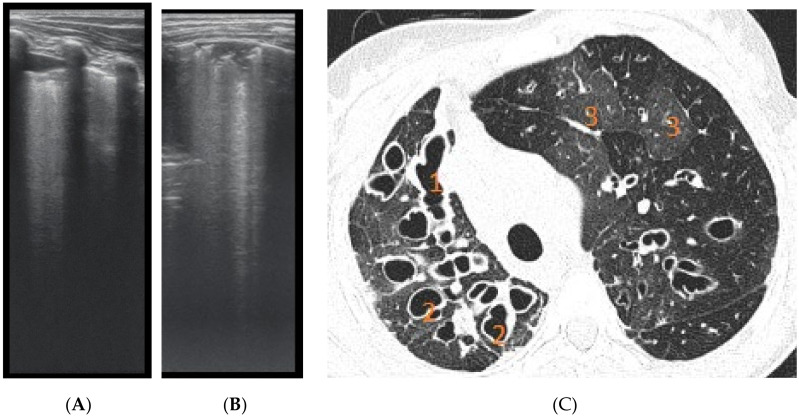
(**A**) LUS: coalescent B lines, erased A profile, loss of aeration, left hemithorax. (**B**) LUS image with subpleural consolidations, coalescent B lines, and left hemithorax. (**C**) CT scan of the same patients: (1) varicose bronchiectasis with middle 1/3 of lung extended and moderate bronchial wall thickening; (2) saccular bronchiectasis with mild and moderate wall thickening; and (3) zones with increased attenuation of pulmonary parenchyma (alveolar infiltrates).

**Figure 8 diagnostics-12-00376-f008:**
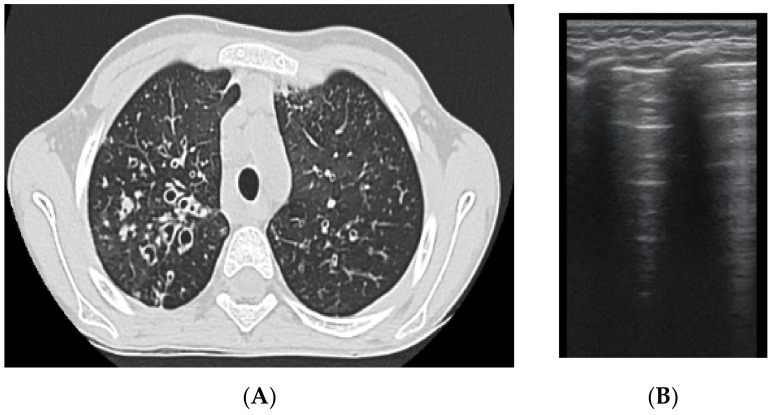
(**A**) CT scan: cylindrical bronchiectasis with mucus plugs (**B**) LUS: A lines, normal LUS aspect, score = 0.

**Figure 9 diagnostics-12-00376-f009:**
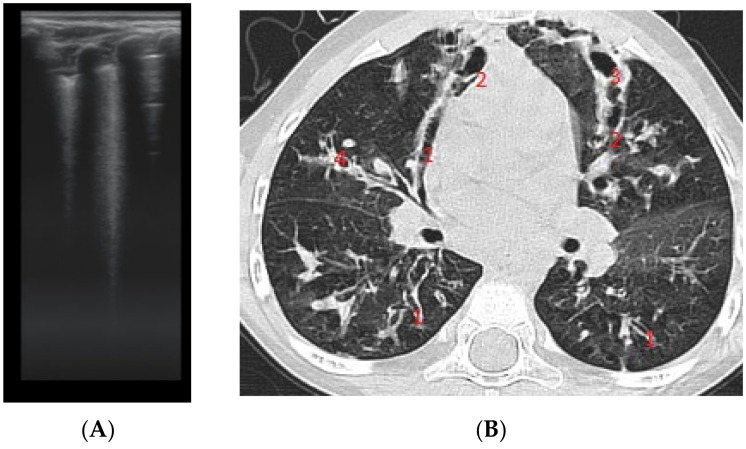
(**A**) LUS: coalescent B lines, loss of A lines. (**B**) CT image: (1) cylindrical bronchiectasis with moderate wall thickening; (2) varicose bronchiectasis; (3) saccular bronchiectasis with moderate wall thickening; and (4) several bronchiectasis with mucus plugging.

**Figure 10 diagnostics-12-00376-f010:**
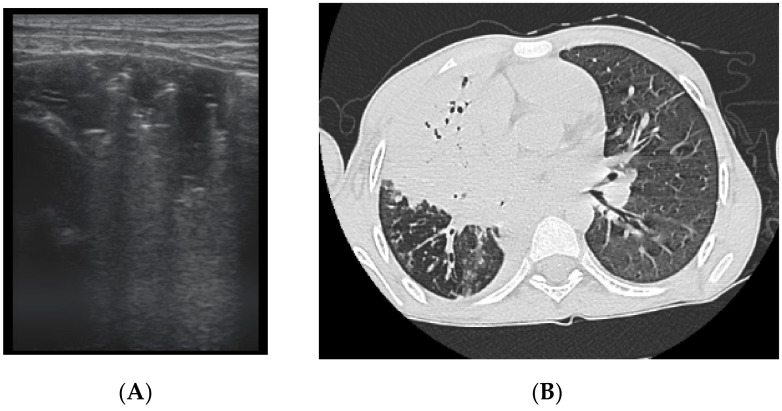
(**A**) LUS image of atelectasis, hypoechoic image with air inside. (**B**) CT scan reveals atelectasis, bronchiectasis, and partial bronchogram.

**Figure 11 diagnostics-12-00376-f011:**
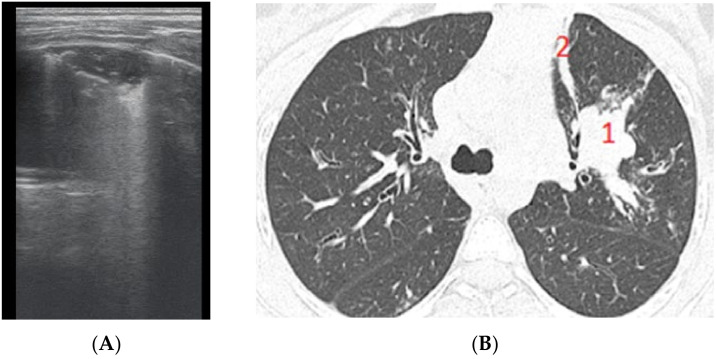
(**A**) LUS image of atelectasis, consolidation without bronchogram. (**B**) CT exam illustrates (1) a peribronchovascular consolidation without air bronchogram, and (2) a lamellar (band) atelectasis.

**Table 1 diagnostics-12-00376-t001:** LUS-CF artefacts score.

LUS Artefact	Lung CF Score
Presence of A lines-normal aspect Distinctive B lines < 3/ic space	0
Distinctive B lines > 3/space or 1 coalescent B line	1
Coalescent B lines > 2/ic space	2
Consolidation < 1 cm	3
Consolidation > 1 cm, with bronchogram	4
Atelectasis/consolidation without bronchogram, > 1 cm	5

**Table 2 diagnostics-12-00376-t002:** Median LUS scores in CT score categories.

	Mild CT Score	Moderate CT Score	Severe CT Score	H	*p*
LUS				39.845	0.000
Median (IQR)	1 (1; 2)	6.5 (4; 11)	15 (12.75; 16.5)		
Mean of rank	16.07	38.14	50.05		

## Data Availability

Data available on request from the corresponding author.

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
