# Peer review of "Lung Ultrasound in Children with Cystic Fibrosis in Comparison with Chest Computed Tomography: A Feasibility Study"

_diagnostics, 2022, doi:10.3390/diagnostics12020376_

Round 1

Reviewer 1 Report

The work presented for the examination is interesting and concerns an issue that is important from a practical point of view. The potential for replacing CT with ultrasound of the lungs in children is very promising, especially in the case of a disease requiring frequent monitoring (CF). A great value of the work is the comparison of the LUS results not only with CT, but also with the results of the functional test - spirometry. At a time when LUS is becoming a more and more common test, data is very much needed to assess the usefulness of this diagnostic method in various specific pathologies. Therefore, the work definitely deserves publication, but I am asking for one editorial amendment - I believe that point 3.2 needs to be redrafted in terms of a clear description of the CF-LS score.

Author Response

Dear Reviewer,

Thank you for considering our paper entitled “Lung ultrasound in cystic fibrosis- a CT comparison study” for resubmission in your journal. We are grateful for your pertinent observation and helpful suggestions. Please find below the answers or completions to your comments.

Q1: The work presented for the examination is interesting and concerns an issue that is important from a practical point of view. The potential for replacing CT with ultrasound of the lungs in children is very promising, especially in the case of a disease requiring frequent monitoring (CF). A great value of the work is the comparison of the LUS results not only with CT, but also with the results of the functional test - spirometry. At a time when LUS is becoming a more and more common test, data is very much needed to assess the usefulness of this diagnostic method in various specific pathologies.

A1: Thank you very much for your consideration for the subject, that we also consider it extremely important for clinical practice.

Q2:Therefore, the work definitely deserves publication, but I am asking for one editorial amendment - I believe that point 3.2 needs to be redrafted in terms of a clear description of the CF-LS score.

A2: It was a great observation, the table from 3.2 was corrected at you notice, we copied it below for reference; thank you for noticing.

LUS Artifact

Lung CF score

Presence of A lines-normal aspect

Distinctive B lines<3 / ic space

0

Distinctive B lines>3/space or 1 coalescent B line

1

Coalescent B lines >2/ic space

2

Consolidation<1cm

3

Consolidation>1 cm, with bronchogram

4

Atelectasis/Consolidation without bronchogram, > 1 cm

5

Sincerely yours,

Authors

Reviewer 2 Report

Abstract: Provides a good overview of the study

Keywords: Fine

Ethical Committee: Addressed.

Title: This should be considered an exploratory study as a first experiment in the field. The title should state so (e.g. view the terms "pilot" or "exploratory" or "feasibility"). Besides, from this title, it is not clear the actual aim of the study at all. So, a possible effective title should be “Diagnostic value of Lung Ultrasound in children with Cystic Fibrosis in comparison with chest computed tomography: a feasibility study”

Introduction: the topic is high-relevant but it is not introduced in a consistent way. A concise and brief description of the role of modified Bhalla score and lung clearance index should be provided.

Materials and methods:  the experience of the radiologist in lung ultrasound and CT imaging, expressed in years, should be added.

Discussion: the authors should provide a comparison with other ultrasound scoring systems developed for cystic fibrosis [10.1371/journal.pone.0215786] [10.1186/s12890-021-01728-8] in terms of imaging findings and results differences. In particular, pleural irregularities were described and analysed in previous studies whereas, they were not included in the present study.  

Minor language typos to be corrected.

Statement: The paper faces a high-relevant topic and it is well-organized. The role of ultrasound for the monitoring of children with cystic fibrosis poses a radiological challenge according to the ionizing radiation dose in the paediatric population. This study explored the feasibility of lung ultrasound in the diagnosis and monitoring of pulmonary changes performing an interesting correlation with CT and clinical data. Nevertheless, minor revisions are needed before acceptance.

Author Response

Dear Reviewer,

Thank you very much for revising the study previously entitled “Lung ultrasound in cystic fibrosis- a CT comparison study”. We are grateful for your pertinent observation and helpful suggestions. Please find below the answers or completions to your comments.

Q1:Title: This should be considered an exploratory study as a first experiment in the field. The title should state so (e.g. view the terms "pilot" or "exploratory" or "feasibility"). Besides, from this title, it is not clear the actual aim of the study at all. So, a possible effective title should be “Diagnostic value of Lung Ultrasound in children with Cystic Fibrosis in comparison with chest computed tomography: a feasibility study”

A1: Thank you for suggestion of changing the title, the new title is: Lung Ultrasound in Children with Cystic Fibrosis in comparison with Chest Computed Tomography: a Feasibility Study. Unfortunately, although very clear, the suggestion of title  “Diagnostic value of Lung Ultrasound in children with Cystic Fibrosis in comparison with chest computed tomography: a feasibility study” is very similar to the study published already : “Diagnostic value of chest ultrasound in children with cystic fibrosis – Pilot study” and we tried to change it for better comprehension.

Q2: Introduction: the topic is high-relevant but it is not introduced in a consistent way. A concise and brief description of the role of modified Bhalla score and lung clearance index should be provided.

A2:  Thank you very much for the idea of high lightening the importance of modified Bhalla score and LCI. Please find below the fragment that was completed in the introduction of the paper:

“Chest CT is the gold standard for structural evaluation of the CF lung disease, and the need of an objective marker led to CT scores that includes estimation of the severity and degree of CF specific features [33].  Bhalla modified score is an accurate and feasible way to assess the severity of CF lung parenchymal disease, having high correlation with lung function [33], severe genotype and chronic Pseudomonas infections [32,33,35]. The modified Bhalla score by Leung A. et al. [35] includes evaluation of presence, severity and extent of bronchiectasis, bronchial wall thickening, mucus plugging, atelectasis/consolidation and air trapping, using 0-3 scale severity [35], simplifying the original Bhalla scale [20].

Even CT scores were used as a surrogate outcome in the evaluation of cystic fibro-sis lung disease, it seems that LCI is comparable with CT in terms of sensitivity. LCI is an indicator of irregular ventilation distribution, that sensibly detects abnormal lung structure early changes in the CF lungs [32,33]. Its practical usefulness consists in the applicability among younger children, that cannot perform spirometry, as only tidal breathing is necessary for performing multiple breath wash-out (MBW), the method through LCI is obtained [32]. Several studies demonstrated that LCI is better correlated with CT scores than spirometry parameters and very sensitive for detection of early changes in CF lung. [32,33], suggesting that “LCI may be even more sensitive than HRCT scanning for detecting lung involvement in CF” [32]. 

Therefore, the choice of comparation of our new LUS-CF score with CT, the gold standard for structural CF changes and LCI, the most accurate lung function parameter, is the correct premise for this feasibility study.

Q3: Materials and methods:  the experience of the radiologist in lung ultrasound and CT imaging, expressed in years, should be added.

A3: The lung ultrasound was performed by an experienced trained pediatric pulmonologist, with 7 years of experience in LUS, blinded to previous CT examinations and images were checked by the senior radiologist, with 8 years expertise in lung ultrasound. 

The images were stored in the workstation, hospital’s PACS system and CDs, and were interpreted by one experienced radiologist with CT competence and 16 years experience 

Q4:Discussion: the authors should provide a comparison with other ultrasound scoring systems developed for cystic fibrosis [10.1371/journal.pone.0215786] [10.1186/s12890-021-01728-8] in terms of imaging findings and results differences. In particular, pleural irregularities were described and analysed in previous studies whereas, they were not included in the present study.  

A4: Please find below the completion discussion regarding reference to other studies;

“The first study that presented the CF lung ultrasound artefacts in cystic fibrosis described the presence of interstitial syndrome, bronchiectasis, alveolar consolidation, and pleural signs but was published only in abstract, by our group [41].  Strzelczuk–Judka reported afterwards the CF-USS(Cystic Fibrosis Ultrasound Score) , that evaluated the presence and extent of pleural irregularities, focal or coalescent ”lung rockets” B lines, subpleural consolidations and pleural fluid, showing a positive correlation with Nor-man Crispin CXR scoring systems (r = 0.52, p = 0.0002) [16]. Like in our study, they acknowledge as an important limitation, the inability of visualization of  the respiratory airway deterioration like bronchiectasis and  mucus plugs.

Peixoto and colleagues reported a score that included A pattern, B pattern, consolidation stratifying patients in A profile, B profile, C profile or mixed profiles com-pared to CT and reported a good correlation between LUS and CT [17]. The presence of pleural effusion was not score (declared as very rare) neither the pleural irregularities (only de-scribing the finding) [17] , similar to our study. The choice of excluding pleural effusion from our LUS-CF score consisted in the fact that pleural effusion is not a specific CF feature.

Hassanzad and colleagues noted in LUS the presence of pleural thickening, atelectasis, air bronchogram, B-line, and consolidation, compared the findings with corresponding   CXR and HRCT, evaluated the diagnostic performance of LUS and CXR for every arti-facts with satisfactory results [40].  Similar to our discoveries, a good diagnostic performance for the detection of consolidation was reported in this paper.

Regarding pleural irregularities, Strzelczuk–Judka noted pleural irregularities similar to bronchiolitis in one patient [16], Peixoto describing, but not quantifying, this feature [17].  We did not take into consideration the presence of the pleural irregularities for this 2016 starting study, because CT did not show a specific corresponding finding, therefore we considered it normal appearance at the time of our CF LUS score development. Neither the others studies did not stated that pleural irregularities or thickening would correspond to a specific modification on CT.

With previous experience of our group on LUS in CF [41] , we noticed that LUS artifacts may quantified different pathological expression. As exemplified in the results of the present paper, B lines can quantify also interstitial inflammation or small bronchiectasis, as confirmed by CT in our study. Similarly, the coalescent B lines can be suggestive for alveolo-interstitial inflammation or mucous plugging with loss of aeration, either bronchial wall ticketing. We observed that subpleural consolidation with absence of A lines quantified small atelectasis but also cystic bronchiectasis filled with mucus. The lack of specificity for mentioned artifacts made us to be cautioned with a statement of LUS specificity for CF lung disease. However, CF-LUS score showed a very good correlation with CT, as mentioned and the great sensitivity of the method to detect parenchymal abnormalities. 

Q5: Minor language typos to be corrected.

A5: English native speaker corrected the manuscript, as recommended.

Sincerely yours,

Authors